



Earth System
Dynamics

# Modelling sea-level fingerprints of glaciated regions with low mantle viscosity

**Alan Bartholet**[1], **Glenn A. Milne**[1], and **Konstantin Latychev**[2]

[1]Department of Earth and Environmental Science, University of Ottawa, Ottawa, K1N 6N5, Canada
[2]Department of Earth and Planetary Sciences, Harvard University, Cambridge, MA 02138, USA

**Correspondence:** Glenn A. Milne (gamilne@uottawa.ca)

**Abstract.** Global patterns of sea-level change – often termed "sea-level fingerprints" – associated with future changes in ice/water mass re-distribution are a key component in generating regional sea-level projections. Calculation of these fingerprints is commonly based on the assumption that the isostatic response of the Earth is dominantly elastic on century timescales. While this assumption is accurate for regions underlain by mantle material with viscosity close to that of global average estimates, recent work focusing on the West Antarctic region has shown that this assumption can lead to significant error where the viscosity is significantly lower than typical global average values. Here, we test this assumption for fingerprints associated with glaciers and ice caps. We compare output from a (1D) elastic Earth model to that of a 3D viscoelastic model that includes low-viscosity mantle in three glaciated regions: Alaska, southwestern Canada, and the southern Andes (Randolph Glacier Inventory (RGI) regions 1, 2, and 17, respectively). This comparison indicates that the error incurred by ignoring the non-elastic response is of the order of 1 mm in most areas (or about 1 % of the barystatic signal) over the 21st century with values reaching the centimetre level in glaciated regions. However, in glaciated regions underlain by low-viscosity mantle, the non-elastic deformation can result in relative sea-level changes with magnitudes of up to several tens of centimetres (or several times the barystatic value). The magnitude and spatial pattern of this non-elastic signal is sensitive to variations in both the projected ice history and regional viscosity structure, indicating the need for loading models with high spatial resolution and improved constraints on regional Earth viscosity structure to accurately simulate sea-level fingerprints in these regions. The anomalously low mantle viscosity in these regions also amplifies the glacial isostatic adjustment signal associated with glacier changes during the 20th century, causing it to be an important (and even dominant) contributor to the modelled relative sea-level changes over the 21st century.

## 1 Introduction

A variety of processes drive changes in the vertical position of the ocean floor and ocean surface (e.g. Church et al., 2013; Milne et al., 2009), and the combination of these processes produces a complex pattern of relative sea-level (RSL) change that varies through time. While the global mean RSL change provides a useful single value which reflects the contribution from climate-related processes, specifically land ice melt and ocean warming, and does represent a reasonable estimate of sea-level change at many coastal locations, various regional processes that produce a strong signal can result in large departures from the global mean value (Church et al., 2013). As a result, predicting future sea-level changes at regional to local scales is challenging, as it requires calculating and summing signals associated with numerous physical processes that have a range of spatial scales and response times (Slangen et al., 2012; 2014; Kopp et al., 2014).

Around the world, glaciers and ice sheets are losing mass and retreating (e.g. Oppenheimer et al., 2019; Vaughan et al., 2013; The IMBIE team, 2018, 2020; Wouters et al., 2019; Zemp et al., 2019). Observations since 1850 CE show that,

on a global scale, the rate of glacier mass loss in the early 21st century is without precedent for the observation period (Zemp et al., 2015). The melting of ice sheets and glaciers produces a spatial pattern of sea-level change due to the resulting solid Earth deformation and changes to the geopotential (Farrell and Clark, 1976). When these changes happen on decadal to centennial timescales, the resulting solid Earth response is assumed to be dominantly elastic; thus, the non-elastic (viscous) contribution is commonly neglected. The modelled spatial patterns in RSL change associated with these short-term changes in ice mass are often termed "sea-level fingerprints" (e.g. Mitrovica et al., 2011). These fingerprints play a central role in projections of regional sea-level change (Church et al., 2013; Oppenheimer et al., 2019; Palmer et al., 2020; Slangen et al., 2012, 2014; Spada, 2017).

The assumption of an insignificant contribution of the non-elastic signal to sea-level fingerprints was recently addressed in a paper focusing on mass loss of the West Antarctic ice sheet (Hay et al., 2017). In this region, the viscosity of the Earth's shallow mantle has been inferred to be as much as several orders of magnitude lower than the global average value (e.g. Whitehouse et al., 2019). Hay et al. (2017) concluded that the viscous component of the response is significant and thus should be included when computing sea-level fingerprints. In this study, we extend this discussion to regions with glaciers that are underlain by low-viscosity mantle. A number of studies have provided evidence that the glaciated regions of Alaska, western Canada and USA, and the southern Andes are located in regions where the sub-lithosphere mantle viscosity has been estimated to be 1–2 orders of magnitude lower than typical global mean values (e.g. Hu and Freymueller, 2019; James et al., 2009; Jin et al., 2017; Richter et al., 2016). The cause of such low viscosity is likely to be related to the presence of plate subduction and the influx of water-rich fluids from the subducting oceanic plate into the overlying mantle (e.g. Brocher et al., 2003). Departures from an elastic response will be relatively large in these regions and so the computed sea-level fingerprints may be in significant error. The primary aim of this work is to quantify the amplitude and spatial extent of the error caused by assuming an elastic Earth response in the three regions mentioned above. In particular, a key goal is to determine if the influence of non-elastic deformation in these low-viscosity regions acts to significantly influence the calculated fingerprints beyond the regions defined by low-viscosity mantle material.

## 2  Methods

Our sea-level projections were generated using a numerical finite-volume formulation of the surface loading process (e.g. Latychev et al., 2005; Hay et al., 2017). This formulation assumes a spherical Maxwell body, discretized using a tetrahedral grid in which the lateral resolution is greatest ($\sim 12$ km)

at the surface of the Earth model and lowest ($\sim 50$ km) at the core–mantle boundary. We solve the sea-level equation using the approach described in Mitrovica and Peltier (1991) but extended to incorporate the influence of Earth rotation on RSL changes (Milne and Mitrovica, 1998; Mitrovica et al., 2005). In order to apply this algorithm, two primary inputs must be defined: a realistic space-time evolution of grounded land ice to force the model and a realistic model of the Earth that defines the interior density and rheology structure to compute the viscoelastic response. These two model inputs are detailed below.

### 2.1  Ice model

In this study we created ice models for each of the 19 first-order regions in the Randolph Glacier Inventory 5.0 (RGI; Pfeffer et al., 2014). The RGI provides the area of glacier extent in each of the regions, and then we apply the region-specific thickness–area scaling function of Huss and Farinotti (2012), which calculates the mean thickness of each glacier in a region as follows:

$$\bar{h} = c S^{\gamma}, \qquad (1)$$

where $\bar{h}$ is the mean thickness, $S$ is the area of the glacier, and $c$ and $\gamma$ are constants specific to each region in the RGI. In order to determine a mass loss history for our ice model for all 19 regions in the RGI we use the decadal Representative Concentration Pathway (RCP) 4.5 projections provided by Huss and Hock (2015) for the period 2010–2100 CE with a net global barystatic (Gregory et al., 2019) sea-level change of 10.8 cm. The barystatic sea-level change for RGI regions 1, 2, and 17 are 1.8, 0.2 and 0.3 cm for RCP4.5, respectively (Huss and Hock, 2015).

Using the decadal mass loss projections, we produced a model of ice extent changes that simulates the vertical thinning of the ice, as well as a crude estimate of lateral retreat as the area of ice cover changes. We iterated over each of the decadal time steps and calculated the amount of uniform ice thickness change (based on areal extent) required to equal the projected sea-level equivalent (SLE) using a tolerance of $\pm 1$ %. We then subtracted this height from the ice thickness distribution of the previous time step and revised the area distribution to account for locations where ice thickness had reduced to zero. We then applied a spatial Gaussian filter to the calculated change in ice extent between successive time steps (using the NumPy 1.16.1 multidimensional Gaussian filter) to spatially smooth the ice thickness distribution. While this did result in some loss of spatial fidelity, it removed the need for anomalously large changes in ice thickness to produce the desired volume changes. This process was applied individually to each of the 19 first-order regions in the RGI. Figure 1a shows ice extent at 2010 and 2100 CE for the RGI regions 1 (Alaska) and 2 (western Canada and USA), and Fig. 1b gives the same results for region 17 (southern Andes).

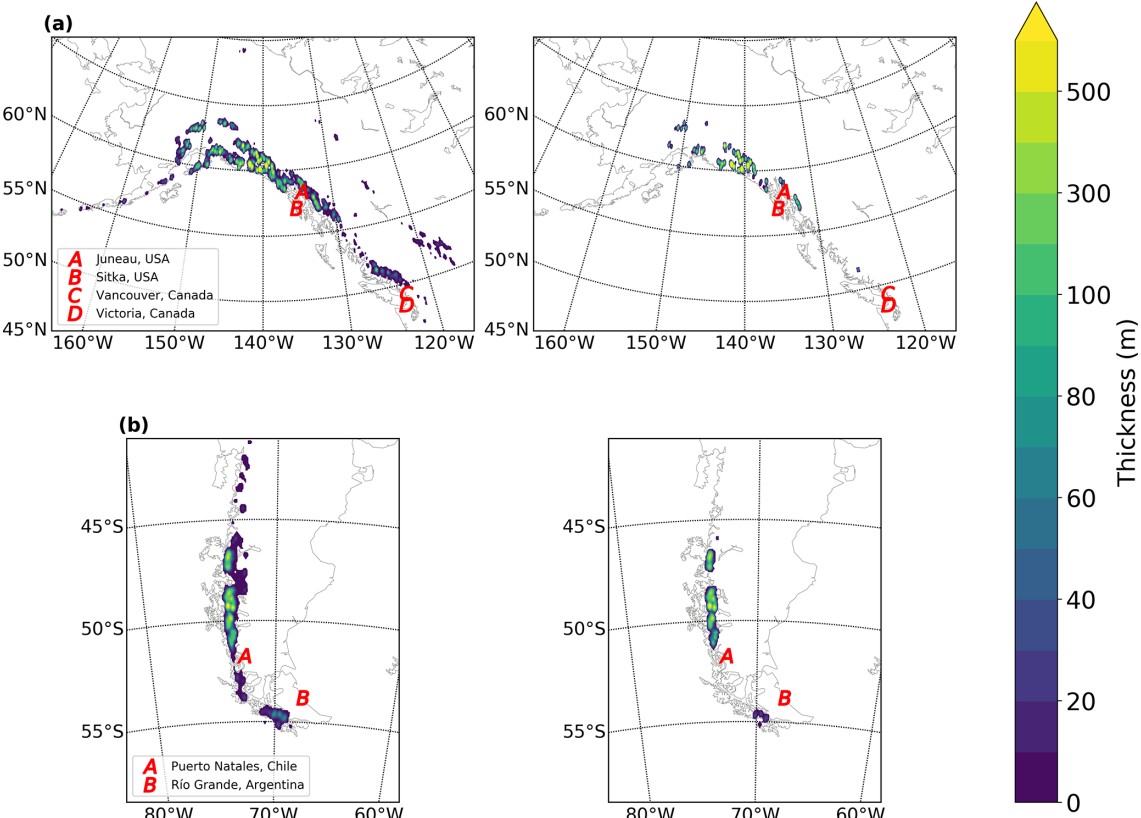

**Figure 1.** Estimated spatial distribution in ice thickness in RGI regions 1 and 2 **(a)** and 17 **(b)** at the beginning (2010 CE, left) and end (2100 CE, right) of the time period considered. The locations of population centres for which relative sea-level curves are calculated (see Figs. 5 and 6) are indicated by the red letters.

While the focus of this study is sea-level fingerprints associated with future changes in glaciers, we also briefly consider the influence of loading changes during the 20th century as a glacial isostatic adjustment (GIA) overprint on the 21st century fingerprint signal. We apply the same methods as described above but use the regional volume change estimates from Marzeion et al. (2015) to determine ice thickness changes going backwards in time from 2010 CE. The total barystatic sea-level changes for RGI regions 1, 2, and 17 for the period from 1902 to 2010 CE are 0.7, 0.3, and 0.2 cm, respectively. In comparison to the 21st century loading model, no lateral changes were incorporated in the 20th century model, as thicknesses generally increased at each time step (going backwards from 2010 CE). That is, the lateral extent remains fixed to that defined in the RGI (5.0).

## 2.2   Earth model

The density and elastic properties of our Earth model are defined using the radial (1D) seismic Preliminary Reference Earth Model (Dziewonski and Anderson, 1981). We note that the influence of lateral variations in elastic and density structure on the computation of sea-level fingerprints has been

shown to be negligible (Mitrovica et al., 2011). Due to large uncertainty in our knowledge of the viscosity structure of the Earth, the radial variation of this structure is most commonly defined by only three parameters: the first is an outer shell of high viscosity ($1 \times 10^{37}$ Pa s), which is used to simulate an elastic outer shell (the lithosphere); the second is an iso-viscous upper mantle region, which extends from the base of the lithosphere to a depth of 670 km; and the third is an iso-viscous lower-mantle region that extends from 670 km to the core–mantle boundary (2885 km). The values used to define the viscosity vary depending on the region detailed below, resulting in an Earth model where the viscosity structure varies not only with depth but laterally as well.

In defining global-scale viscosity structure, we assign a lithospheric thickness of 96 km, an upper-mantle viscosity of $5 \times 10^{20}$ Pa s, and a lower-mantle viscosity of $10^{22}$ Pa s. While there is considerable uncertainty in our knowledge of global average viscosity structure, most of this uncertainty relates to that of the lower mantle (e.g. Mitrovica and Forte, 2004; Lambeck et al., 2014). The values we use for lithospheric thickness and upper-mantle viscosity are broadly compatible with those from recent analyses of global GIA data sets (Lambeck et al., 2014; Peltier, 2004), and the value

we use for lower-mantle viscosity represents a middle ground between these recent estimates. Given the short time period of our model simulations ($\sim 100\,\mathrm{yr}$), the use of other global average viscosity structures could be substituted without significantly impacting the results, as the component of non-elastic deformation is small for viscosity values typically inferred in global GIA analyses. The regional viscosity structure we adopt is the more important aspect of our Earth model, as this is anomalously low in RGI regions 1, 2, and 17.

For RGI region 1 (Alaska), a number of studies have estimated the regional viscosity structure (e.g. Larsen et al., 2005; Sato et al., 2011; Jin et al., 2017; Hu and Freymueller, 2019). All of these studies estimate a relatively thin lithosphere elastic thickness, averaging around 50 km but with uncertainty of a few tens of kilometres and low viscosity values in the shallow upper mantle ranging between middle $10^{18}\,\mathrm{Pa\,s}$ to low $10^{19}\,\mathrm{Pa\,s}$. In a recent analysis, Jin et al. (2017) used measurements from Ice Cloud and Land Elevation Satellite (ICESat), global positioning system (GPS), and Gravity Recovery and Climate Experiment (GRACE) to estimate Earth model parameters. By isolating the signal due to past ice loading, they concluded on a best fit three-layer Earth model consisting of a lithospheric (elastic) thickness of 60 km, a 110 km thick asthenosphere with a viscosity of $2 \times 10^{19}\,\mathrm{Pa\,s}$, and a sub-asthenosphere mantle with a viscosity of $4 \times 10^{20}\,\mathrm{Pa\,s}$. A second recent study (Hu and Freymueller, 2019) also used vertical land motion rates from GPS to constrain a regional, depth-dependent viscosity structure. They estimated lithosphere thickness to be 55 km, and the viscosity and thickness of the asthenosphere to be $3 \times 10^{19}\,\mathrm{Pa\,s}$ and 230 km, respectively, but noted significant trade off in these parameter values.

The relatively good agreement between these studies gives some confidence in choosing parameters. We adopted the values of Jin et al. (2017) for this study (those from Hu and Freymueller (2019) were published after the completion of our modelling) and extended their sub-asthenosphere region (with a viscosity of $4 \times 10^{20}\,\mathrm{Pa\,s}$) to the bottom of the upper mantle (670 km); below this depth, values associated with the global background model are the default. The lateral extent of these viscosity values at the model Earth surface is shown in red in Fig. 2a. In order to constrain the lateral extent of the low-viscosity region, we define a surface area that is roughly similar to the region studied by Jin et al. (2017). Note that the lateral extent of this region decreases with depth as it is projected radially downwards.

For RGI region 2 (western Canada and USA), we are interested only in the area adjacent to southwestern British Columbia, as this is where GIA studies have inferred low-viscosity values. James et al. (2009) concluded that RSL observations from Vancouver Island can be fit equally well across a wide range of asthenosphere thicknesses and viscosities. The Earth model with the lowest viscosity consisted of a lithospheric (elastic) thickness of 60 km, a 140 km thick asthenosphere with a viscosity of $3 \times 10^{18}\,\mathrm{Pa\,s}$, and a sub-

asthenosphere mantle with a viscosity of $4 \times 10^{20}\,\mathrm{Pa\,s}$. These results are supported by a more recent study that considered sea-level observations from a larger area in southwestern British Columbia (Yousefi et al., 2018), and thus we adopt the values from James et al. (2009) to define the regional lithosphere thickness and upper-mantle viscosity structure. The lateral extent of this region at the model Earth surface is shown in green in Fig. 2a.

In the southern Andes region (RGI area 17), a number of studies have inferred the presence of low-viscosity material (e.g. Ivins and James, 1999, 2004; Lange et al., 2014; Richter et al., 2016) that likely resides in the mantle wedge between the subducting plate and the base of the lithosphere (Klemann et al., 2007). In all of these studies, the estimated lithosphere elastic thickness is relatively thin ($\sim 30\,\mathrm{km}$) and asthenosphere viscosity low (order $10^{18}\,\mathrm{Pa\,s}$). We adopted results from the most recent of the above-listed analyses: Richter et al. (2016), who used observations from 43 geodetic Global Navigation Satellite System (GNSS) sites distributed over the Southern Patagonian Ice Field to analyse vertical and horizontal velocities of present-day crustal deformation. By applying an ice-load history that assumes a moderate present-day glacial mass loss, with slightly higher than present-day mass loss immediately following the Little Ice Age (LIA) maximum, Richter et al. (2016) decided on a preferred Earth model consisting of a 36.5 km thick lithosphere and a sub-lithosphere mantle with a viscosity of $1.6 \times 10^{18}\,\mathrm{Pa\,s}$. They applied a half-space Earth model and thus provide no constraint on the depth extent of the low-viscosity asthenosphere. We place the lower asthenosphere boundary at 150 km with viscosity values of our reference model below this depth. The lateral extent of this low-viscosity region is shown in Fig. 2b.

## 3    Results and discussion

Our goal is to quantify the signal of the non-elastic response to sea-level fingerprints computed for the three RGI regions introduced above. Therefore, in the following we focus on comparing results from our 3D viscoelastic Earth model to those computed using an elastic Earth model in which properties vary only with depth.

Figure 3a shows the global sea-level fingerprint for the non-linear mass loss results of Huss and Hock (2015) as applied to all regions of the RGI, assuming that Earth deformation is entirely elastic. As is conventional, the fingerprint shows the total sea-level change between the start and end of the study period (in this case, 2010 to 2100 CE). The pattern of sea-level change is typical in that it shows a sea-level fall near the sources of the ice mass loss and a sea-level rise in the far field (e.g. Mitrovica et al., 2001). Using the same ice loading model, results for the 3D viscoelastic Earth model are shown in Fig. 3b. At the global scale, comparison between the results in Fig. 3a and b shows that the differences

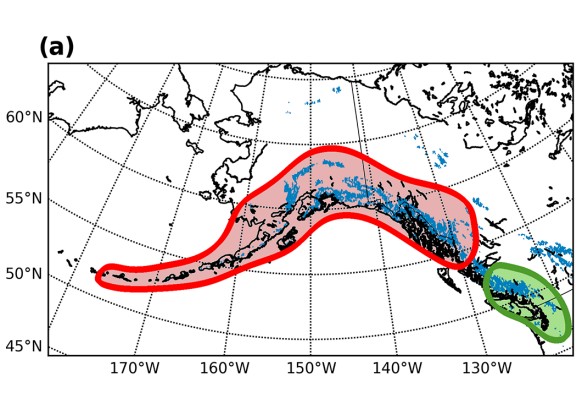

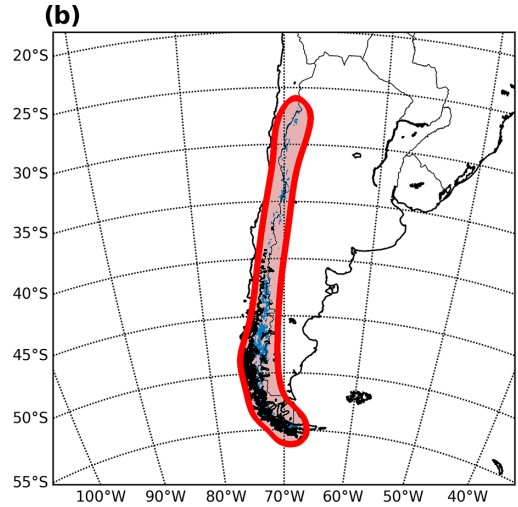

**Figure 2.** Surface lateral extent of the regions for which the underlying Earth structure (lithospheric thickness and sub-lithosphere viscosity profile) deviate from the adopted global values: panel **(a)** shows the extent for RGI regions 1 (red) and 2 (green), while panel **(b)** shows the extent for region 17.

are small; therefore, we subtract the elastic results from the viscoelastic results to isolate the difference (Fig. 3c). At locations located away from glaciated areas, the difference between the viscoelastic and elastic Earth models is negligible (generally of the order of 1 mm at 2100 CE). In glaciated regions where the viscosity structure is that of the global model (i.e. all RGI regions except 1, 2, and 17), differences approach the 1 cm level by 2100 CE (or about 10 % of the barystatic sea-level change), indicating that the assumption of an elastic Earth is relatively accurate even in near-field regions when the viscosity of the underlying mantle is close to global average values. However, in areas where anomalously low viscosity values exist (Fig. 2), the difference in sea-level change can be several times larger than the barystatic value (10.8 cm) due to the faster response of the low-viscosity mantle. Thus, the error introduced by considering only the elastic solid Earth response is spatially restricted and significantly exceeds the centimetre level only in the vicinity of the low-viscosity regions. The remainder of this section will focus on the signal in these near-field regions only.

Sea-level predictions for the regions with underlying low-viscosity mantle are shown in Fig. 4. The results for Alaska and western Canada and USA (Fig. 4a–c) indicate that the spatial pattern associated with the viscoelastic signal is markedly different from that for the elastic Earth model. Note that, for ease of interpretation, the predictions shown in Fig. 4 consider only the RSL change associated with ice mass loss in the respective RGI regions (1 and 2 in Fig. 4a–c and 17 in Fig. 4d–f), hence the difference between the results in Figs. 3 and 4. The influence of ice changes in other RGI regions will cause an almost uniform signal over each of the areas shown in Fig. 4, with amplitude close to the global barystatic value,

and thus the spatial patterns (RSL gradients) will not be significantly affected by this omission.

Inspection of Fig. 4 indicates that the spatial pattern associated with the viscoelastic signal is markedly different to that of the elastic Earth model. When an elastic model is used, the near-field RSL signal is entirely negative, reflecting a lowering of the geoid and uplift of the solid Earth. In this case, the entire region experiences uplift as shown in Fig. S1 in the Supplement. In the viscoelastic case, the low-viscosity values reduce the Maxwell time of the material such that the non-elastic component of deformation becomes significant after a few decades. As a result, areas of subsidence peripheral to the ice-covered regions are predicted (Figs. 4b, e and S1b, e). These areas of net subsidence (so-called "peripheral bulges") are a characteristic feature of the GIA response on millennial timescales (e.g. Peltier, 1974; Clark et al., 1978; Whitehouse, 2018) and reflect the isostatic signal of a thin elastic lithosphere overlying a viscoelastic mantle within which the non-elastic component of deformation is significant.

Focusing first on the results for RGI regions 1 and 2 (Fig. 4a–c), in areas where ice has thinned or disappeared (Fig. 1), the difference in RSL (Fig. 4c) shows a larger fall, whereas peripheral to this area, the RSL fall is lower (with some regions showing a small RSL rise by 2100 CE – blue bands peripheral to the central uplifting area in Fig. 4b and e). Because of the considerably lower mantle viscosity in these regions, the solid Earth responds faster than it otherwise would have over the same time period. As a result, the areas shaded in red (Fig. 4c) show a greater sea-level fall compared to the elastic case due to the additional uplift of the solid Earth surface associated with non-elastic deformation. The peripheral areas showing a reduced RSL fall or subtle RSL

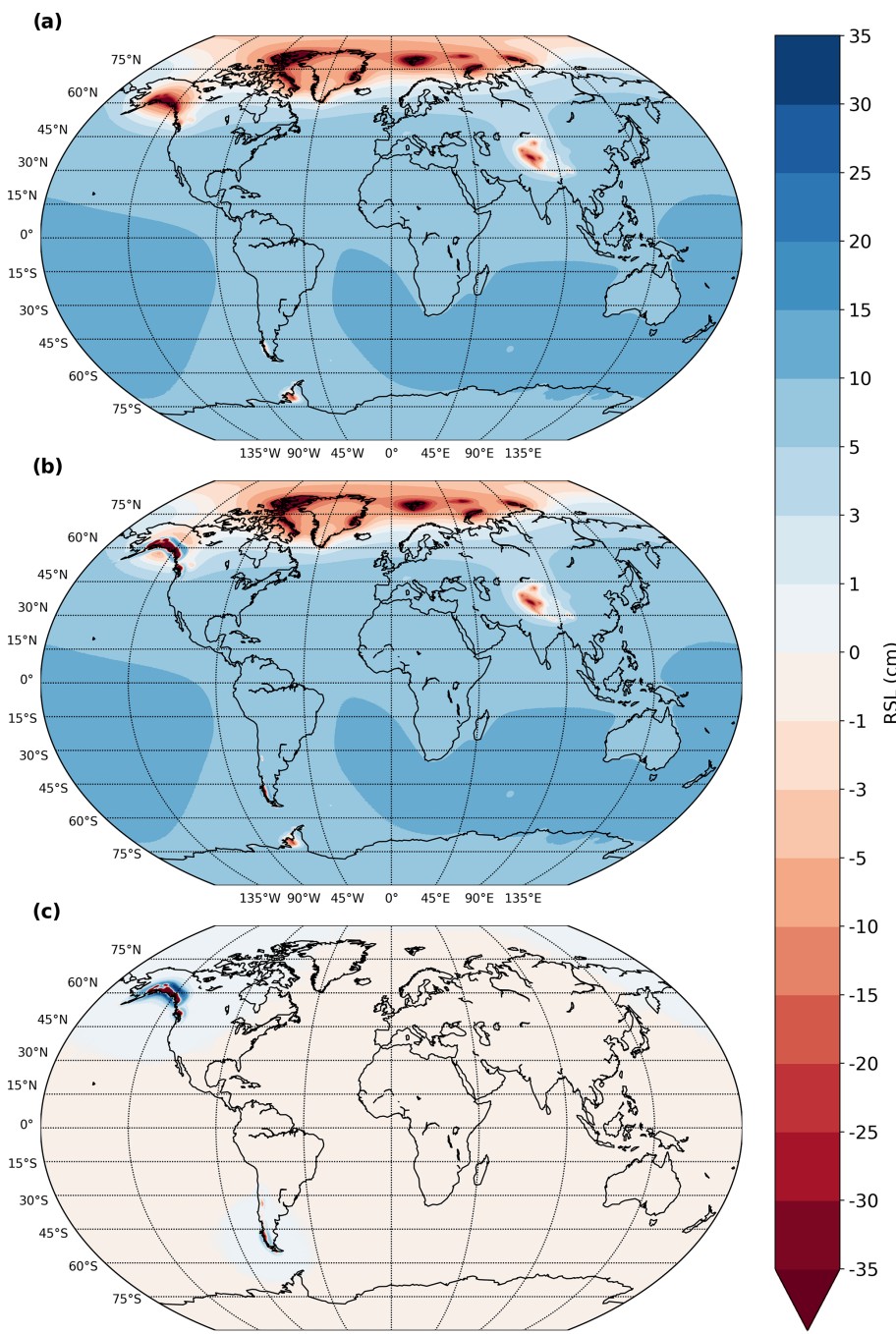

**Figure 3.** Calculated sea-level fingerprints for estimated changes in global glacier distribution from 2010 to 2100 CE for **(a)** a 1D (spherically symmetric) elastic Earth model and **(b)** a 3D viscoelastic Earth model with low-viscosity regions located as indicated in Fig. 2. **(c)** The difference between the viscoelastic and elastic results (i.e. **b** minus **a**).

rise relate to non-elastic deformation that governs the formation of peripheral bulges. The results in Fig. 4c indicate that the error made by assuming an elastic Earth response can exceed several tens of centimetres and be positive or negative depending on the location. The differences between the viscoelastic and elastic model runs are dominated by vertical

land motion, which is shown in Fig. S1 using the same format as Fig. 4. The sea surface component of RSL is, in general, smaller than the land motion signal, but this is site and time dependent (see Fig. 5 and related discussion below).

Results for the Southern Andes (Figs. 4d–f and S1d–f) are similar in that the non-elastic component of deformation in

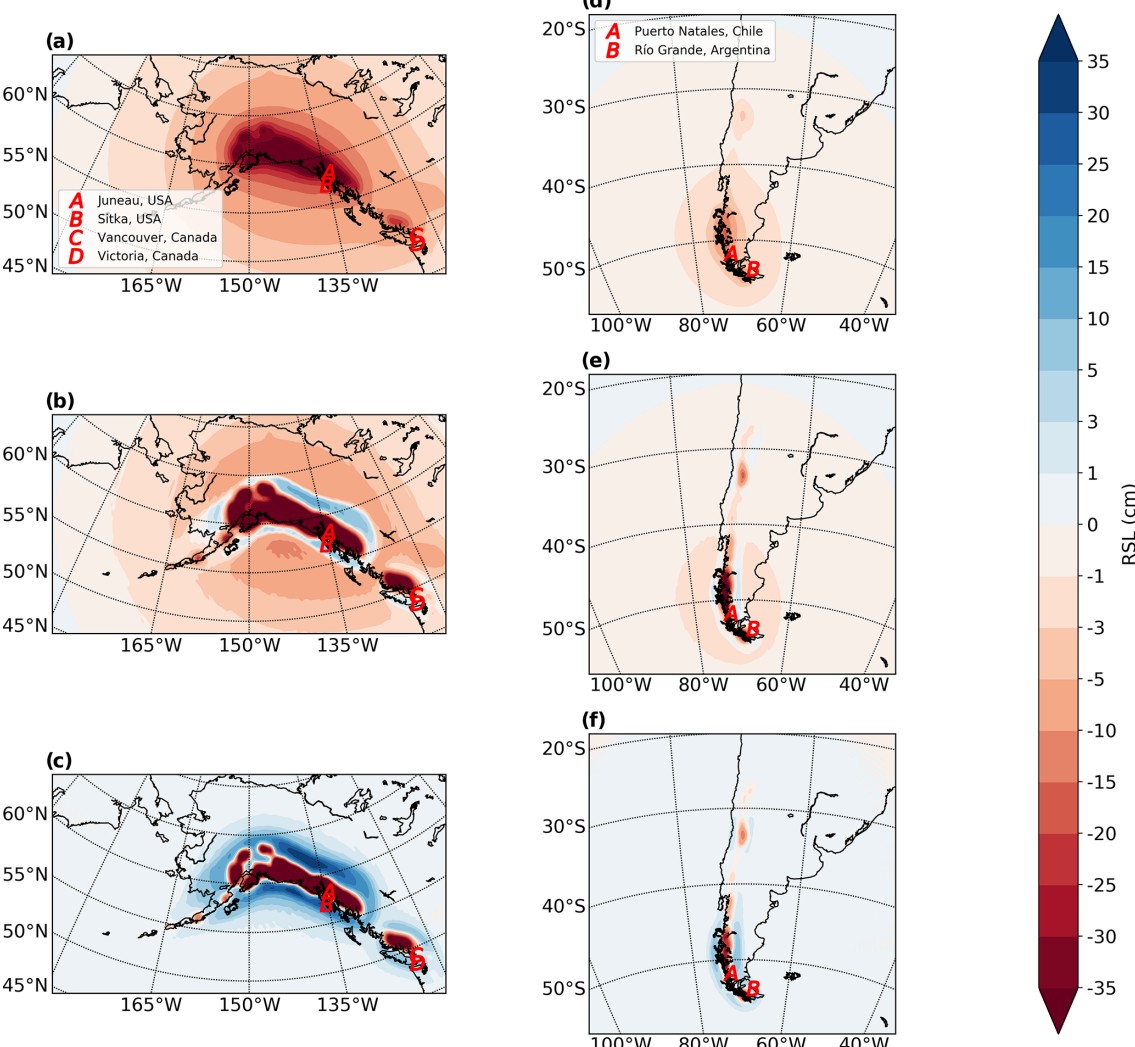

**Figure 4.** Calculated sea-level fingerprints for estimated changes in regional glacier distributions for RGI regions 1 and 2 (left) and region 17 (right). The different frames show results for **(a, d)** a 1D (spherically symmetric) elastic Earth model and **(b, e)** a 3D viscoelastic Earth model with low-viscosity regions located as indicated in Fig. 2. The results in **(c)** and **(f)** show the differences between the viscoelastic and elastic results, respectively (i.e. **b** minus **a** and **e** minus **d**). Note that these results do not include the sea-level signal associated with ice mass changes from outside of the RGI regions shown. The locations of population centres for which relative sea-level curves are calculated (see Figs. 5 and 6) are indicated by the red letters.

this region results in a more rapid sea-level fall in areas of ice thinning or retreat. The RSL differences (compared to the elastic case) are not as large as they are for Alaska because the amplitude of ice mass loss is lower (Fig. 1), but the differences still reach values of $\sim 10$ cm and thus are large relative to the barystatic signal (10.8 cm). The peripheral region of subsidence is less well developed compared to that for RGI region 1. This is due to the different amplitude and geometry of ice mass loss (Fig. 1) and the difference in the viscosity structure between the two regions.

The spatial patterns shown in Fig. 4 are complemented by model output of time series for six different towns or cities (population ranging from $\sim 10\,000$ to several million

inhabitants) in Fig. 5. These particular locations (indicated in Figs. 1 and 4) were chosen to illustrate the range of RSL signals evident in near-field areas. RSL time series for RGI regions 1 and 2 are shown for Juneau and Sitka (Region 1), Vancouver and Victoria (Region 2), and Puerto Natales and Río Grande (Region 17). Time series for both the elastic and viscoelastic cases are based on a 10-year discretization of the ice thickness model as described in Sect. 2.1. The 1D viscoelastic results (blue lines) are based on the reference (global) viscoelastic model. At all locations shown in Fig. 5, the difference between these results and those based on an elastic model (red lines) are at the centimetre level or less, which is consistent with the results in Fig. 1c for glaciated

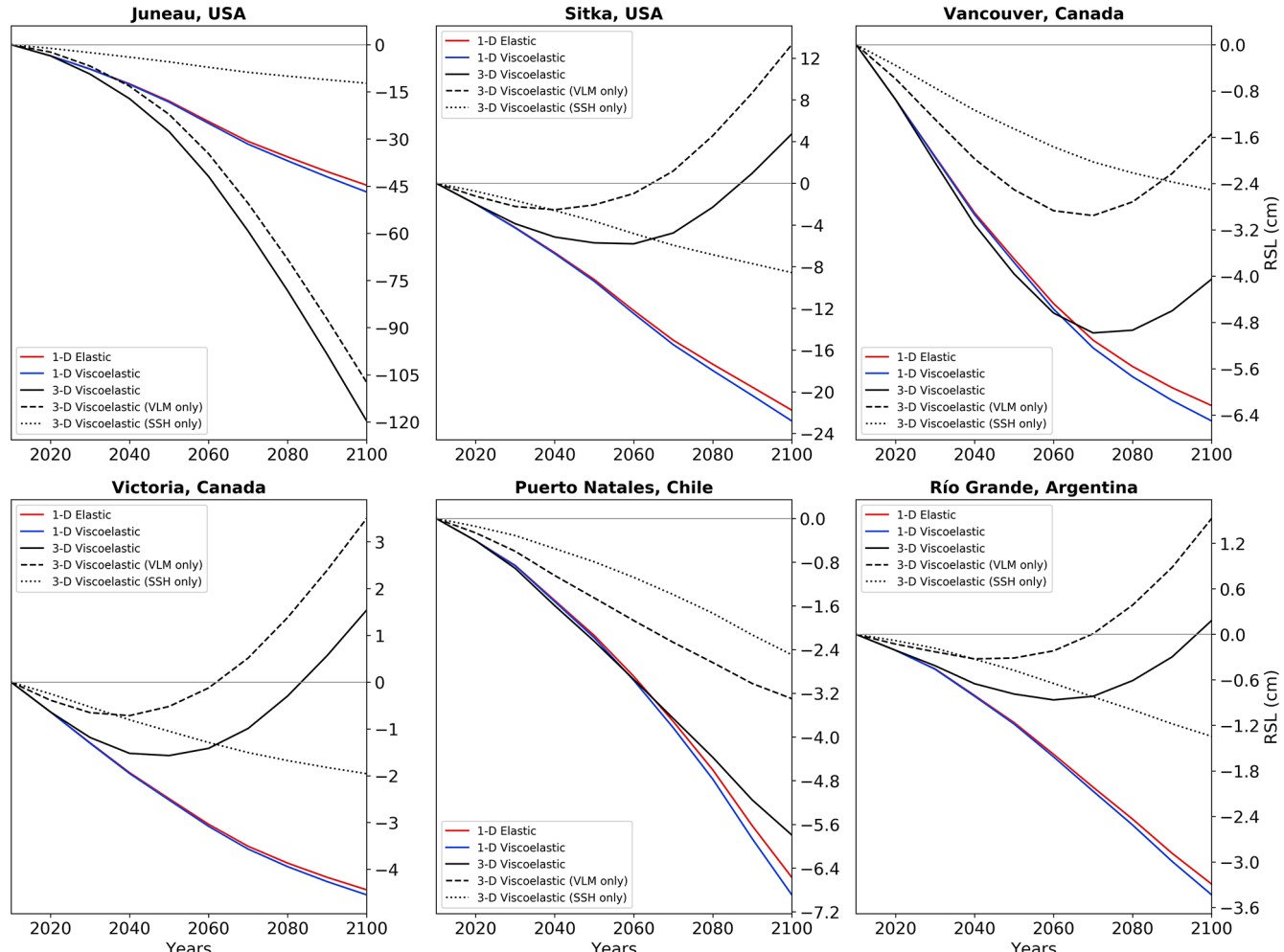

**Figure 5.** Calculated RSL curves showing the time variation of the spatial patterns in Fig. 4 at the locations indicated in Figs. 1 and 4. The results for a 1D viscoelastic model using our reference global viscosity profile (Sect. 2.2) are also included. The contributions of vertical land motion (VLM) and sea-surface height (SSH) change to RSL are also shown for the 3D viscoelastic Earth model. As for Fig. 4, these results do not include the sea-level signal associated with ice mass changes from outside the respective RGI regions.

regions not underlain by anomalously low-viscosity mantle. The 3D viscoelastic results (black lines) include the low-viscosity regions illustrated in Fig. 2 and described in Sect. 2.2. At Juneau, USA, a sea-level fall is predicted for both the elastic and 3D viscoelastic results, with the latter showing a much greater fall (by ∼ 74 cm). The dashed and dashed–dotted black lines show the component signals for the 3D viscoelastic case, and these indicate that the vertical land motion (VLM) contribution to RSL dominates over sea-surface height (SSH) change at this site. Clearly, the application of an elastic Earth model greatly underpredicts the sea-level fall at this location. The predicted RSL curves for Puerto Natales are similar in that a sea-level fall is also shown; however, at this location the 3D viscoelastic model predicts a smaller fall (by ∼ 1 cm) compared to the elastic model. The lower RSL amplitudes at this location reflect the

smaller ice mass changes and the location of the settlement relative to the area of major mass loss.

At Sitka, Vancouver, Victoria, and Río Grande, results for the viscoelastic Earth model give a RSL response that transitions from a fall to a rise. This is due to the more complex spatial pattern of the predicted response when a viscoelastic Earth model with anomalously low-viscosity material is applied. As noted above, the GIA response is characterized by uplift in regions of mass loss and subsidence in some of the areas peripheral to the glaciers (Fig. S1b and e). Looking at the results for these four locations, the non-monotonic nature of the RSL response is governed by that of the VLM; the SSH contribution is primarily that of a sea-level fall associated with the reduction in ice mass, resulting in a diminishing gravitational pull on the surrounding ocean. This fall in SSH offsets some of the sea-level rise caused by the VLM in these locations. The non-monotonic shape of the RSL curve has the

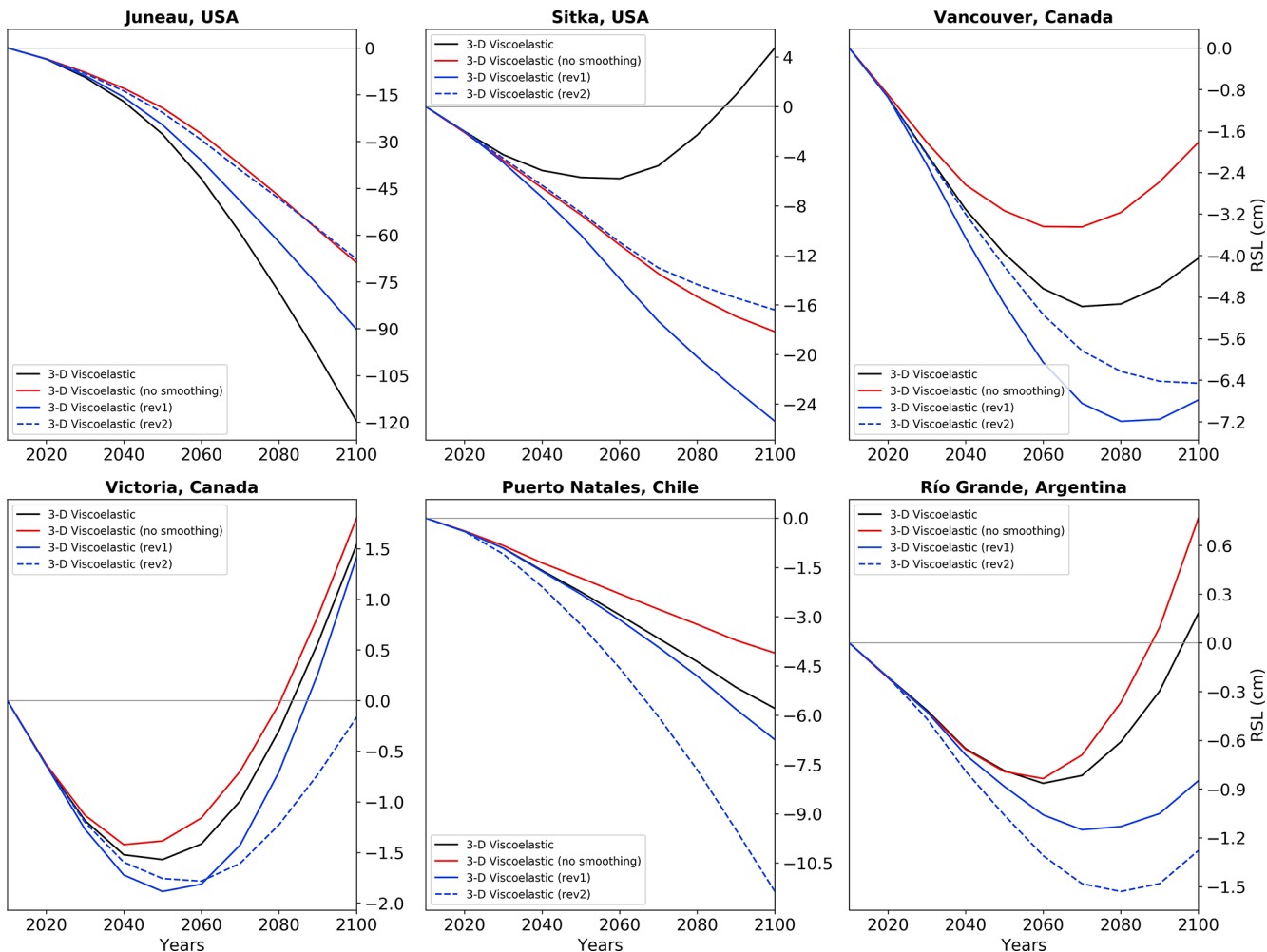

**Figure 6.** Calculated RSL curves showing the sensitivity of model output to changes in input parameters. The 3D viscoelastic results (black lines) are calculated using the same model parameters used to determine the results shown in Figs. 4 and 5. The coloured lines in each frame show the influence of changing the regional ice model (red line) or two different aspects of the regional Earth viscosity model (blue lines). As for Fig. 5, these results do not include the sea-level signal associated with ice mass changes from outside the respective RGI regions.

net effect of producing a relatively small RSL change over the 21st century – at all four sites exhibiting this behaviour, the amplitude of RSL change is no more than a few centimetres. As a consequence, the final difference between the elastic and viscoelastic curves at 2100 CE is also relatively small, except at Sitka where the RSL rise of ∼ 4 cm for the viscoelastic model compares to a large fall of ∼ 22 cm for the elastic case.

The results in Figs. 4 and 5 are based on only one realization of viscosity structure and one estimate of ice thickness changes in each of the three regions. A small number of additional model runs were performed to give a crude indication of the sensitivity of the results to changes in these primary model inputs. Figure 6 shows modelled RSL curves at the same locations in Fig. 5 wherein one aspect of the model input was changed (red and blue lines). We show results for the case where the ice loading history was not

smoothed (Fig. S2), and thus the ice thickness changes are generally larger and more spatially restricted compared to the smoothed case (Fig. 1), particularly at later times in the 21st century when the lateral ice extent has significantly reduced. This change in the spatial fidelity of the ice distribution leads to a significant change in the predicted RSL curves at all locations, but this is particularly evident at Juneau and Sitka. While the largest change is at Juneau, with a decrease in predicted RSL fall by ∼ 50 cm, the change at Sitka is notable due to the change in sign. A spatial map of the change in RSL (2100 CE relative to 2010 CE) is shown in Fig. S3a, which can be compared to the results in Fig. 4b. In general, the differences relative to the elastic model are smaller, though they are still at a magnitude of several decimetres in some locations (such as Juneau). Also of note is that the more localized ice loading does not lead to the prediction of RSL rise during the 21st century in the Alaska region

(this is not the case for the other two RGI regions considered). The greater sensitivity of the Alaskan results to this change in the ice model most likely reflects the larger mass changes in this region. Given that there is considerable sensitivity to this aspect of the model input at all low-viscosity locations (Fig. S4a and d), we conclude that the application of a global-scale GIA model with uniform spatial resolution is not an optimal approach to model the near-field isostatic response to the detailed changes in ice distribution illustrated in Figs. 1 and S2. As a result of the limited spatial resolution of our GIA model, Iceland, which is known to be underlain by low-viscosity mantle, was not included in the analysis. Further work should focus on the use of regional (Cartesian) models or global models with non-uniform grids (e.g. Larour et al., 2017) or nested, high-resolution regional grids (e.g. Goldberg et al., 2016) to more accurately capture the detailed loading history and associated isostatic response in these low-viscosity RGI regions.

The blue lines in Fig. 6 indicate the sensitivity of our model results to changes in the Earth model viscosity structure. The solid blue lines show the results for model runs in which the lateral extent of the low-viscosity regions was significantly extended (Fig. S5). Comparison of the black and (solid) blue lines indicates that the results are sensitive to this aspect of the Earth viscosity model, with the greatest sensitivity found in Alaska where the RSL differences (relative to the original 3D Earth model) reach several tens of centimetres. At the other locations, the differences are generally less than a few centimetres (Figs. 6 and S4b and e). In the final sensitivity test, we kept the lateral extent of the low-viscosity areas the same (Fig. 2) but changed the depth of the bottom of this region to coincide with the 670 km depth seismic discontinuity. While this is not a particularly realistic scenario, it serves to make a preliminary assessment of the impact of this model parameter on the output. The results of this change are shown by the dashed blue lines in Fig. 6 and the maps in panels c and f of Figs. S3 and S4. These changes to the viscosity structure in each region also significantly impact the predicted RSL changes. Again, the largest sensitivity is evident in Alaska, with amplitudes of several decimetres at both Juno and Sitka. At the other locations, the sensitivity tends to be within the range $\pm 5$ cm in the western Canada and US region and $\pm 10$ cm in the southern Andes region. Clearly, changes to both the lateral and depth extent of the low-viscosity region have a significant impact on the model output; therefore, we recommended the use of more realistic Earth models that consider additional constraints such as regional seismic velocity models and/or viscosity structure associated with subduction (e.g. Austermann et al., 2013; Klemann et al., 2007; Yousefi et al., 2021).

The large amplitude of the non-elastic signal on century timescales in low-viscosity regions indicates that application of an elastic model can result in significant error in the calculated sea-level fingerprint. A potentially important implication of this result is that the isostatic response to mass loss

changes during the 20th century could be a significant contributor to the RSL response during the 21st century. The importance of this earlier loading signal is evident in the large contemporary uplift rates measured in the regions considered (e.g. Hu and Freymueller, 2019; Richter et al., 2016). We note that this signal is generally not considered to be a sea-level fingerprint as it is due to past ice mass changes. Instead, it is considered as part of the GIA component of the regional sea-level projections (e.g. Slangen et al., 2014). The RSL contribution of our 20th century ice loading model (1902–2010 CE; see Section 2.1) is shown in Fig. S6 (dashed black lines) along with the contribution from the load model shown in Fig. 1 (solid black lines; these are the same as the solid black lines in Fig. 5). As expected, the 20th century signal is monotonic as there is no active loading after 2010 CE. The amplitude at the six sites considered ranges from $\sim 10$ to $\sim 2$ cm (between 2010–2100 CE). The sign of the change due to 20th century loading at each location is compatible with the rate of change of the RSL fingerprints towards the end of the 21st century (e.g. sites showing a positive RSL fingerprint trend are where the 20th century signal is a RSL rise from 2010 to 2100 CE). At some sites where the fingerprint signal is highly non-monotonic and thus results in a small net RSL change (Victoria, Rio Grande), the 20th century loading signal dominates the total changes from 2010–2100 CE. Figure S7b and e show the regional influence of 20th century loading on RSL at 2100 CE relative to 2010 CE. Overall, our results indicate that the 20th century signal can be at the 5–10 cm level and thus should be considered when generating regional RSL projections in these low-viscosity areas.

While this study assumes a Maxwell rheology, it is possible that, on the relatively short timescales considered here, significant departures from this simple rheological model may occur. These departures could take the form of a transient component of the non-elastic response (e.g. Yuen et al., 1986; Pollitz, 2005), in which the viscosity increases with time or a power-law response that is often associated with relatively large deviatoric stress (e.g. Wu and Wang, 2008; van der Wal et al., 2013) for which the effective viscosity would increase as stress levels relax. The significance of these more complex rheological models in low-viscosity regions would be a natural extension of this analysis.

## 4 Conclusions

Sea-level fingerprints are an integral aspect of calculating regional variations in future sea-level change. Calculation of these fingerprints commonly assumes that the isostatic response of the Earth is elastic on century time scales. Here we tested this assumption by comparing output from a (1D) elastic Earth model to that of a 3D viscoelastic model, which includes low-viscosity mantle in three glaciated regions: Alaska, southwestern Canada, and the southern Andes (RGI regions 1, 2, and 17, respectively). This comparison indicates

that the error incurred by ignoring the non-elastic response is generally of the order of 1 mm over the 21st century but can reach magnitudes of up to several tens of centimetres when proximal to glaciated regions overlying anomalously low-viscosity mantle. Our model results show significant sensitivity to variations in the input ice distribution history and regional viscosity structure. Given this, a logical extension of this work would apply models with high spatial resolution to adequately capture the ice load changes and incorporate more constraints on regional 3D viscosity structure (e.g. slab geometry or constraints from seismic imaging). We conclude that sea-level fingerprints on elastic Earth models are accurate in most areas but can be in error by an amount several times the global barystatic value (10.8 cm here) in glaciated regions with shallow mantle viscosity that is several orders of magnitude less than that of typical global average values. Furthermore, the low mantle viscosity in these regions amplifies the GIA signal associated with glacier changes during the 20th century, resulting in this signal being an important (even dominant) contributor to the modelled RSL change over the 21st century.

**Code availability.** Information on the software used and how to access it can be obtained from the corresponding author upon request.

**Data availability.** Model output from this work is available from the corresponding author upon request.

**Supplement.** The supplement related to this article is available online at: https://doi.org/10.5194/esd-12-1-2021-supplement.

**Author contributions.** AB performed the research and led the writing of the paper. GAM and KL advised AB in performing the research and contributed to writing the paper.

**Competing interests.** The authors declare that they have no conflict of interest.

**Disclaimer.** Publisher's note: Copernicus Publications remains neutral with regard to jurisdictional claims in published maps and institutional affiliations.

**Acknowledgements.** We thank Jeff Freymueller, Volker Klemann, and an anonymous reviewer for constructive reviews that led to improvements in this work. We acknowledge Matt King for bringing the idea explored in this study to our attention.

**Financial support.** This research has been supported by the Natural Sciences and Engineering Research Council of Canada (grant no. RGPIN-2018-06355).

**Review statement.** This paper was edited by Gerrit Lohmann and reviewed by Jeff Freymueller, Volker Klemann, and one anonymous referee.

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
