# Peer review of "Modelling sea-level fingerprints of glaciated regions with low mantle viscosity"

_Earth System Dynamics, 2020_

## Referee Comment (RC1) · Volker Klemann (Referee) · 30 Oct 2020

Dear Gerrit,

The authors discuss the impact of low viscosity spots in the asthenosphere, associated with tectonic activity, on the sea-level predictions due to glacial changes between 2020 and 2100 CE. While the study is fairly well structured and written, I ended up with a number of concerns regarding their conclusions.

1. From the given figures, I got the impression that the impact of viscoelasticity is important only in regions of low viscosity coinciding with ice-mass loss, whereas the global sea-level fingerprints are merely affected. This aspect is from my point of view the most important result. The phrase "This comparison indicates that the error incurred

by ignoring the non-elastic response is generally less than 1 cm over the 21st century but can reach magnitudes of up to several 10s of centimetres in low viscosity areas" does not reflect this finding properly. Inside the manuscript the authors use stronger expressions.

2. The authors state, that higher resolved load distributions are demanded for reconstructing the spatial pattern of the sea-level fingerprints in the surrounding of such glacial changes. This is expectable due to the thinner lithosphere and viscous response considered in these regions. But a sensitivity study undermining this conclusion is missing. Futhermore, they smoothened their forcing using a Gaussian filter, so reducing lateral variability.

3. The importance of structural features of the asthenosphere are mentioned in the conclusions but not repeated in the abstract as a finding. The authors cite Austermann et al. 2013 and Klemann et al. 2007, but do not deliberate about whether earth structure or load distribution impacts the derived pattern more.

My suggestion is, the authors should discuss Point 2 in more detail, may be by presenting a proper sensitivity study with different details regarding the load distribution. A discussion of Point 3, would be great, but this might be bejond this study. But in this case they should relate to such a future extension. They should also compare their results to a 1D ve earth model in order to prove the reliability to compare their 3D results with those of an elastic model.

Regarding the setup of the manuscript, I must confess that I am not a native speaker. Nevertheless, I had the impressions, that some statements can be expressed more precisely and, at some places, the discussion can be sharpened.

Some details: The authors should care that all abbreviations are explained on first use and should care about the common use of hyphenation. Further things I placed in the annotated manuscript I attached as a supplement.

Best regards, Volker

Please also note the supplement to this comment:
https://esd.copernicus.org/preprints/esd-2020-72/esd-2020-72-RC1-supplement.pdf

---

## Referee Comment (RC2) · Anonymous Referee #2 · 27 Nov 2020

Formal review of manuscript "Modelling sea-level fingerprints of glaciated regions with low mantle viscosity" by Alan Bartholet et al, published as open discussion paper in ESD.

The authors build a 3D viscoelastic model to study the effect of low-viscosity regions in glaciated areas on sea level fingerprints between present day and the end of the century.

They focus on two regions, namely Alaska-SW Canada and the southern Andes, and find that including viscoelastic relaxation in the upper mantle produces large near-field effects, in comparison with results obtained by modelling a purely elastic earth.

The paper is nicely written, clearly and thoroughly explained, and it addresses an issue

that, though well known to the GIA (and to part of the geodetic) community, has not yet been explicitly discussed. There has been a number of studies about viscoelastic effects of recent ice melt (mainly post-LIA) in several regions worldwide (Alaska, Patagonia, Antarctica, Iceland), but when it comes to sea level fingerprints and especially their use for sea level projections, most studies are still based on the elastic approximation, following the work published by Mitrovica and colleagues almost 20 years ago.

The authors show both the importance of local viscosity structures in the near field and the practical validity of the elastic approximation in the far filed, which are both very interesting and useful result.

Hence, I find this work very timely and I strongly recommend its publication in ESD.

Apart from a few minor comments listed below, I have only one somehow major concern.

As the authors explicitly discuss, one of the main differences with respect to an elastic model is the presence of side lobes (or peripheral budges) that move vertically in opposite direction with respect to the areas directly below the ice load. Even though those lobes can only be expected to appear next to the uplifting region, while comparing Figure 4b with Figure 2a I cannot help but noticing that the shape of the subsiding region very closely resembles the shape of the low-viscosity region. That makes me wonder how Figure 4 and 5 would look like, if the low-viscosity region were wider (which I think is a plausible assumption). I understand that the main purpose of this paper is a proof of concept, and the authors already discuss many limitations of their study. Nonetheless, I think that especially the results shown in Figure 5 are very specific and should better be supported by some minimal sensitivity study. A relatively simple option would be to model an end member, i.e., a 1D-model with the same vertical stratification as the center of the low-viscosity zone. That would provide a reasonable lower bound to the effect of local viscosity structures.

Minor comments:

l.11: my -> by (typo).

l.46: mantle -> shallow mantle.

l.53: could you add a short explanation of why plate subduction would change the viscosity structure? It might not be obvious to all readers.

l.62: the sentence is redundant, if layer thickness is already mentioned in the previous line.

l.86: is it 2100 CE or 2090 CE? The caption of Figure 1 seems to suggest the latter.

l.217-218: I wonder if it is realistic not to differentiate between asthenosphere and upper mantle. I understand that this is the model obtained in the reference paper, still it seems a little effort to introduce some reasonable discontinuity.

Figure 4 seems to have a wrong caption (the panel labels in the text do not match the actual panels in the figure).

Figure 5: in all panels, please add a thin horizontal line to indicate RSL=0.

---

## Referee Comment (RC3) · Jeff Freymueller (Referee) · 1 Dec 2020

This paper compares the global and local impacts of 3D viscosity structure on the computation of RSL fingerprints for 21st century ice loss. The basic outline of the results in the low viscosity regions will be no surprise to anyone who has worked on regional GIA models within the low-viscosity regions, but I think that these impacts have not been broadly appreciated among all researchers in GIA. Also, this is the first study that I know of that has made a global quantitative comparison between 1D and 3D models for this purpose, so their evaluation of the worldwide impact (or lack of impact in this case) is new information. Overall, it is good to have a careful quantitative comparison, and I think this paper will have a valuable impact on the field.

[Figure]

The biggest question I have raised by the discussion in lines 235-247. When does the model computation start, and does it include in any way the viscoelastic effect of the 20th century ice loss? It is not explicitly stated, but it appears that the model run starts in 2010, and I suspect that the assumption is that there was no mass change before 2010 (again, this was not explicitly stated, but is implied). That is actually a substantial problem for the featured areas of Alaska and Patagonia, because the viscoelastic effects of the 20th century ice loss will be substantial. I don't think that this invalidates the comparison between the 1D and 3D models in general – if anything, the ongoing viscoelastic response to the 20th century changes will amplify the differences in the near field – and I don't think that this will impact their conclusion that the impacts of the low viscosity regions are essentially local. But it will complicate the discussion that centers around Figure 5. In particular, the position of the hinge line will be affected, so while the trends in the hinge line migration will be largely accurate (especially in the later time periods of the model when the pre-2010 changes are less important as they will be a few relaxation times in the past), any statements about the position of the hinge line may not be accurate without an adequate model for the pre-2010 changes. In particular, the statements about the changes in sign of the RSL trend at Sitka and to a lesser extent Vancouver, Victoria and Rio Grande may or may not be correct here – they depend on the magnitude and sign of the response to the 20th century loss.

I don't see any easy way to deal with this aside from re-running all of the 3D models (for the 1D models the viscosity is so high these changes hardly matter). They could incorporate information from previous modeling studies for Alaska and Patagonia, but those model predictions are only available for the present and thus only define the trend at 2010, not the response over the whole 2010-2100 interval). As a first order approximation it may be good enough to run the 2010-2100 model forward in time to 2210 (with no further changes after 2100), and then add a re-scaled and time-shifted version of this to approximate the 20th century effects. It may also be enough to avoid saying "transitions from a fall to a rise", and instead refer to more positive and more negative RSL rates.

Section 2.1, Ice Model. Can you add something about how this 21st century load differs from the 20th century changes? This should be feasible in the case of Alaska (e.g., Berthier et al., 2010, Nature Geosciences), and Patagonia, which should be sufficient examples. For the world as a whole, the total projected mass loss can be compared with existing global estimates from mountain glaciers. I think it would be adequate to give a percentage change for each region – there will be finer-scale spatial detail changes but they are not important here.

Minor points

Abstract, line 17. The differences 1 cm and 10s of cm don't mean very much without the context of the size of the signal, especially given that the ice load model is a future projection. I suggest at a minimum comparing these to the barystatic sea level rise (10.8 cm) for better context.

Line 39. Add "assumed to be" before "dominantly elastic". This assumption is fine in many cases, although not in the examples this paper focuses upon.

Line 52. Spelling correction: Freymueller

Line 62. This sentence largely repeats the previous one.

Lines 163-168. Same comment as on the Abstract above. In the case of the far-field regions, I wonder if "<1 cm" is the best way to describe it, as it looks on the figure like «1 cm might apply for large areas. I suggest computing a simple statistic like the median absolute deviation for the regions that they define as far-field in panel c, and reporting this quantitative measure.

Line 192. While "greater" and "less than" are mathematically correct, I suggest using the wording "more negative" and "more positive" instead. A similar wording is used a few sentences later. One reason to suggest this change is that it discourages a potential misinterpretation that a reader might make about the absolute value of the RSL change. (I suggest this because my own brain briefly interpreted "less than" as

implying "closer to zero" before I caught myself and realized that the signal in question was already negative.

Line 207. Delete the comma after "change"

Line 265. Spelling correction: Hu, not Huy

Line 274. Spelling correction: van der Wal

Line 279. Do you mean "viscoelastic" rather than "viscosity elastic"?

Lines 287-291. Clarify here whether you incorporated the 20th century changes in your models. My reading is that you did not.

Figure 1. It is really hard to see the colors on the thickness changes. I suggest using a much thinner black line for the coastline, and/or less detail applied for the coastline. Make sure that the resolution of the figures is good enough that readers can zoom in and see the thickness change projections.

Figure 2. The regions are a bit arbitrary here, but I think they are basically defensible. I infer from the text that this was a (former?) students' project and thus not likely to be easily tweaked, but I don't see any clear reason to have a region of high viscosity between the red and green regions in Figure 2a. This will only matter locally and does not impact the main conclusions, so I don't object if it is not easy to change this and it is left as it is.

Figure 5. Some of these curves will be substantially affected by the inclusion of the 20th century ice loss, while others will have only minor impacts. See comments above.

---

## Author Comment (AC1) · 7 Jan 2021

Please see attached supplement.

Please also note the supplement to this comment:
https://esd.copernicus.org/preprints/esd-2020-72/esd-2020-72-AC1-supplement.pdf

---

## Author Comment (AC2) · 12 Jan 2021

Reply to all referees' comments (see attached supplementary file).

Please also note the supplement to this comment:
https://esd.copernicus.org/preprints/esd-2020-72/esd-2020-72-AC2-supplement.pdf

---

## Author Comment (AC3) · 12 Jan 2021

Dear Reviewers,

We appreciate the time and effort you put into reviewing our manuscript and are pleased that, overall, you find our results of interest and a useful contribution to the literature on modelling sea-level fingerprints. Each of you have some reservations and so have requested additional model runs to be performed. As detailed below, we generally agree with the concerns you have raised and describe how the manuscript will be revised to address these. However, many of the additional model runs are on-going and so more time is required to implement the suggested changes.

In the following, we focus only on your comments that require action on our part. The original text from each review is shown in quotation marks and our responses in bold, italicised font. We hope that you find our responses appropriate. Overall, the manuscript will be significantly improved by implementing the changes described below.

Best Regards,

Glenn Milne (corresponding author)

**Reviewer 1 (Dr. Klemann)**

1) "From the given figures, I got the impression that the impact of viscoelasticity is important only in regions of low viscosity coinciding with ice-mass loss, whereas the global sea-level fingerprints are merely affected. This aspect is from my point of view the most important result. The phrase "This comparison indicates that the error incurred by ignoring the non-elastic response is generally less than 1 cm over the 21st century but can reach magnitudes of up to several 10s of centimetres in low viscosity areas" does not reflect this finding properly. Inside the manuscript the authors use stronger expressions."

***Using the detailed comments provided by Dr. Klemann (in an appended pdf document), we have revised the text in a number of places to clarify that for the majority of locations, the application of elastic Earth models to calculate sea-level fingerprints is accurate to within 1 cm over the century time scale considered in our analysis. A significant error (> 5 cm) is predicted only in glaciated regions underlain by anomalously low mantle material. We have also followed the suggestion of Reviewer 3 (Dr. Freymueller; in his minor points) to provide a simple statistical measure of the absolute difference between the viscoelastic and elastic model runs (Fig. 3c) in far-field regions.***

2) "The authors state, that higher resolved load distributions are demanded for reconstructing the spatial pattern of the sea-level fingerprints in the surrounding of such glacial changes. This is expectable due to the thinner lithosphere and viscous response considered in these regions. But a sensitivity study undermining this conclusion is missing. Futhermore, they smoothened their forcing using a Gaussian filter, so reducing lateral variability."

*We will generate model runs with no smoothing applied and compare the results to those with smoothing applied (shown in original Figs 4 & 5). This will provide at least a basic indication of the sensitivity of our results to this aspect of the modelling.*

3) "The importance of structural features of the asthenosphere are mentioned in the conclusions but not repeated in the abstract as a finding. The authors cite Austermann et al. 2013 and Klemann et al. 2007, but do not deliberate about whether earth structure or load distribution impacts the derived pattern more. My suggestion is, the authors should discuss Point 2 in more detail, may be by presenting a proper sensitivity study with different details regarding the load distribution."

*See response to above comment. As for the impact of Earth structure, our aim in this work is to determine if fingerprints are significantly impacted at the global scale by large regional departures from typical, global average viscosity values. Since this is not the case as the impact is only significant within the glaciated regions characterised by anomalously low viscosity, we did not pursue the consideration of more complex viscosity models. It would be more effective to perform this analysis at the regional scale, where the departures from an elastic signal are large and thus worthy of further exploration.*

4) "A discussion of Point 3, would be great, but this might be bejond this study. But in this case they should relate to such a future extension. They should also compare their results to a 1D ve earth model in order to prove the reliability to compare their 3D results with those of an elastic model."

*As noted in our response to comment 3, considering more complex Earth structure models (e.g. with explicit slab-like features) is beyond the scope of this study. We have included model output for a 1-D visco-elastic Earth model (revised Figure 5 and related discussion) as suggested.*

5) "Regarding the setup of the manuscript, I must confess that I am not a native speaker. Nevertheless, I had the impressions, that some statements can be expressed more precisely and, at some places, the discussion can be sharpened."

*A number of revisions to the text have been made to accommodate the suggestions and criticisms detailed in the appended pdf document. We thank Dr. Klemann for taking the time to provide such detailed feedback.*

**Reviewer 2**

1) "Apart from a few minor comments listed below, I have only one somehow major concern. As the authors explicitly discuss, one of the main differences with respect to an elastic model is the presence of

side lobes (or peripheral budges) that move vertically in opposite direction with respect to the areas directly below the ice load. Even though those lobes can only be expected to appear next to the uplifting region, while comparing Figure 4b with Figure 2a I cannot help but noticing that the shape of the subsiding region closely resembles the shape of the low-viscosity region. That makes me wonder how Figure 4 and 5 would look like, if the low-viscosity region were wider (which I think is a plausible assumption). I understand that the main purpose of this paper is a proof concept, and the authors already discuss many limitations of their study. Nonetheless, I think that especially the results shown in Figure 5 are very specific and should better be supported by some minimal sensitivity study. A relatively simple option would be to model an end member, i.e., a 1D-model with the same vertical stratification as the center of the low-viscosity zone. That would provide a reasonable lower bound to the effect of local viscosity structures."

***We agree that a small sensitivity analysis along the lines suggested would improve the rigour of our study. To this end, we will generate new model output based on low viscosity regions that are significantly expanded laterally compared to those shown in Fig. 2. Comparing the results of these runs to those of the original runs will indicate sensitivity of the residual signal (i.e. 3D viscoelastic minus elastic) to this aspect of the model set up. We have also explored the impact of changing the depth to the bottom of the low-viscosity region by extending the based of the asthenosphere layer to the bottom of the upper mantle in Alaska and placing the depth of the low viscosity region in the Southern Andes at a value more consistent with the base of an asthenospheric layer (200 km).***

Minor comments:

***These have all been addressed.***

"l.11: my -> by (typo).

l.46: mantle -> shallow mantle.

l.53: could you add a short explanation of why plate subduction would change the viscosity structure? It might not be obvious to all readers.

l.62: the sentence is redundant, if layer thickness is already mentioned in the previous line.

l.86: is it 2100 CE or 2090 CE? The caption of Figure 1 seems to suggest the latter.

l.217-218: I wonder if it is realistic not to differentiate between asthenosphere and upper mantle. I understand that this is the model obtained in the reference paper, still it seems a little effort to introduce some reasonable discontinuity."

***As discussed in our response to comment #1, we have now included a model run that does isolate the low viscosity zone to a shallow asthenospheric layer.***

"Figure 4 seems to have a wrong caption (the panel labels in the text do not match the actual panels in the figure).

Figure 5: in all panels, please add a thin horizontal line to indicate RSL=0."

**Reviewer 3 (Dr. Freymueller)**

1) "The biggest question I have raised by the discussion in lines 235-247. When does the model computation start, and does it include in any way the viscoelastic effect of the 20th century ice loss? It is not explicitly stated, but it appears that the model run starts in 2010, and I suspect that the assumption is that there was no mass change before2010 (again, this was not explicitly stated, but is implied). That is actually a substantial problem for the featured areas of Alaska and Patagonia, because the viscoelastic effects of the 20th century ice loss will be substantial."

*We agree that the pre-2010 loading (particularly that during the 20th century) will be important, as outlined in lines 262-266. We chose not to include this signal in our analysis as sea-level fingerprints, by convention, define the spatial pattern of sea-level change to future ice mass loss. The signal to past loading changes are included in the so-called 'GIA' component, which is a separate analysis (even though the underlying processes are the same). However, this distinction is often not clearly stated and we agree that our study is a good example of a case where it would be logical to combine past and future loading. One issue in doing this, however, it the large uncertainly in defining 20th century loading changes in this region.*

"I don't think that this invalidates the comparison between the 1D and 3D models in general – if anything, the ongoing viscoelastic response to the 20th century changes will amplify the differences in the near field – and I don't think that this will impact their conclusion that the impacts of the low viscosity regions are essentially local. But it will complicate the discussion that centers around Figure 5. In particular, the position of the hinge line will be affected, so while the trends in the hinge line migration will be largely accurate (especially in the later time periods of the model when the pre-2010 changes are less important as they will be a few relaxation times in the past), any statements about the position of the hinge line may not be accurate without an adequate model for the pre-2010 changes. In particular, the statements about the changes in sign of the RSL trend at Sitka and to a lesser extent Vancouver, Victoria and Rio Grande may or may not be correct here – they depend on the magnitude and sign of the response to the 20th century loss. I don't see any easy way to deal with this aside from re-running all of the 3D models (for the 1D models the viscosity is so high these changes hardly matter). They could incorporate information from previous modeling studies for Alaska and Patagonia, but those model predictions are only available for the present and thus only define the trend at 2010, not the response over the whole 2010-2100 interval). As a first order approximation it may be good enough to run the 2010-2100 model forward in time to 2210 (with no further changes after 2100), and then add a re-scaled and time-shifted version of this to approximate the 20th century effects. It may also be enough to avoid saying "transitions from a fall to a rise", and instead refer to more positive and more negative RSL rates."

*We thank Dr. Freymueller for his suggestion. However, we have decided to take a different approach which, although still crude as it assumes no changes in lateral ice extent, has been applied in past analyses. The approach to defining the 20th century changes is the same as that we applied for the 21st century except that we assume the lateral extent of the ice is fixed and equal to that in the RGI (5.0) database. Thus, at each time step (10 year increments), there is a uniform change in ice thickness that is calculated based on this extent and the*

*volume changes given in Marzeion et al. (2015). Details are provided in the revised manuscript.*

*Again, since the focus of our work is the impact of low viscosity regions on computing fingerprints (associated with future ice mass loss) considering the GIA signal due to past loading changes is not strictly within our remit. However, by including a 20$^{th}$ century signal we demonstrate the importance of glacier changes during this period on the GIA contribution to future sea-level change in these regions. Clearly, more work is necessary to better define the 20$^{th}$ century loading changes and examine the importance of loading changes in the centuries prior to 1900 CE. As we indicated in the original manuscript (lines 266-268), GIA models previously used in making regional sea-level projections do not accurately capture these recent glacier mass changes.*

2) "Section 2.1, Ice Model. Can you add something about how this 21st century load differs from the 20th century changes? This should be feasible in the case of Alaska (e.g., Berthier et al., 2010, Nature Geosciences), and Patagonia, which should be sufficient examples. For the world as a whole, the total projected mass loss can be compared with existing global estimates from mountain glaciers. I think it would be adequate to give a percentage change for each region – there will be finer-scale spatial detail changes but they are not important here."

*As part of our response to #1, we have provided volume change estimates for our three study regions as taken from Marzeion et al. (2015) and compared these to the results in Huss and Hock (2015) for future ice volume loss in the same three regions.*

Minor points

*Note that the minor points (e.g. misspellings) made below have been addressed even though this is not explicitly stated.*

"Abstract, line 17. The differences 1 cm and 10s of cm don't mean very much without the context of the size of the signal, especially given that the ice load model is a future projection. I suggest at a minimum comparing these to the barystatic sea level rise (10.8 cm) for better context."

*Agree - change made.*

"Line 39. Add "assumed to be" before "dominantly elastic". This assumption is fine in many cases, although not in the examples this paper focuses upon."

*Change made.*

"Line 52. Spelling correction: Freymueller

Line 62. This sentence largely repeats the previous one."

*Sentence deleted.*

"Lines 163-168. Same comment as on the Abstract above. In the case of the far-field regions, I wonder if "<1 cm" is the best way to describe it, as it looks on the figure like «1 cm might apply for large areas. I suggest computing a simple statistic like the median absolute deviation for the regions that they define as far-field in panel c, and reporting this quantitative measure."

*Implemented for regions bounded by*

Line 192. While "greater" and "less than" are mathematically correct, I suggest using the wording "more negative" and "more positive" instead. A similar wording is used a few sentences later. One reason to suggest this change is that it discourages a potential misinterpretation that a reader might make about the absolute value of the RSL change. (I suggest this because my own brain briefly interpreted "less than" as implying "closer to zero" before I caught myself and realized that the signal in question was already negative.

*Suggestion implemented for the region bounded by latitudes 45 N and 45 S.*

"Line 207. Delete the comma after "change"

Line 265. Spelling correction: Hu, not Huy

Line 274. Spelling correction: van der Wal

Line 279. Do you mean "viscoelastic" rather than "viscosity elastic"?

Lines 287-291. Clarify here whether you incorporated the 20th century changes in your models. My reading is that you did not."

*See response to main comment #1.*

"Figure 1. It is really hard to see the colors on the thickness changes. I suggest using a much thinner black line for the coastline, and/or less detail applied for the coastline. Make sure that the resolution of the figures is good enough that readers can zoom in and see the thickness change projections."

*Suggestion implemented.*

"Figure 2. The regions are a bit arbitrary here, but I think they are basically defensible. I infer from the text that this was a (former?) students' project and thus not likely to easily tweaked, but I don't see any clear reason to have a region of high viscosity between the red and green regions in Figure 2a. This will only matter locally and does not impact the main conclusions, so I don't object if it is not easy to change this and it is left as it is."

*While we agree that making two distinct viscosity regions for RGI regions 1 & 2 is awkward given their proximity and common tectonic environment, we did this because the viscosities inferred in these areas are different. A sentence has been added to make this point.*

"Figure 5. Some of these curves will be substantially affected by the inclusion of the 20th century ice loss, while others will have only minor impacts. See comments above."

*See response to main comment #1.*

---

## Author Response (AR1)

Dear Reviewers,

We appreciate the time and effort you put into reviewing our manuscript and are pleased that, overall, you find our results of interest and a useful contribution to the literature on modelling sea-level fingerprints. Each of you have some reservations and so have requested additional model runs to be performed. As detailed below, we have incorporated a (small) sensitivity study into the analysis based on your feedback. We have also included results on the contribution of 20[th] century loading to RSL during the 21[st] century. Overall, the paper has been extended by ~10%, an additional figure has been added to the main text (Fig. 6) and a new supplementary section added that includes seven figures. It has also been re-written in some parts to improve clarity.

In the following, we focus only on your comments that require action on our part. The original text from each review is shown in quotation marks and our responses in bold, italicised font. We hope that you find our responses appropriate. Thanks again for your input.

Best Regards,

Glenn Milne (corresponding author)

**Reviewer 1 (Dr. Klemann)**

1) "From the given figures, I got the impression that the impact of viscoelasticity is important only in regions of low viscosity coinciding with ice-mass loss, whereas the global sea-level fingerprints are merely affected. This aspect is from my point of view the most important result. The phrase "This comparison indicates that the error incurred by ignoring the non-elastic response is generally less than 1 cm over the 21st century but can reach magnitudes of up to several 10s of centimetres in low viscosity areas" does not reflect this finding properly. Inside the manuscript the authors use stronger expressions."

***Using the detailed comments provided by Dr. Klemann (in an appended pdf document), we have revised the text in a number of places to clarify that for the majority of locations, the application of elastic Earth models to calculate sea-level fingerprints is accurate in most locations to order 1 mm (i.e. 1% of the barystatic signal) over the 21[st] century (see revised Abstract and Conclusions).***

2) "The authors state, that higher resolved load distributions are demanded for reconstructing the spatial pattern of the sea-level fingerprints in the surrounding of such glacial changes. This is expectable due to the thinner lithosphere and viscous response considered in these regions. But a sensitivity study undermining this conclusion is missing. Futhermore, they smoothened their forcing using a Gaussian filter, so reducing lateral variability."

***We included results for a model run where the ice model was not smoothed and compare the results to those with smoothing applied (Figs 6, S4 & S5). The differences are significant and***

*thus support our conclusion that future studies should use models with non-uniform/nested grids (with enhanced resolution in near-field regions).*

3) "The importance of structural features of the asthenosphere are mentioned in the conclusions but not repeated in the abstract as a finding. The authors cite Austermann et al. 2013 and Klemann et al. 2007, but do not deliberate about whether earth structure or load distribution impacts the derived pattern more. My suggestion is, the authors should discuss Point 2 in more detail, may be by presenting a proper sensitivity study with different details regarding the load distribution."

*A small sensitivity analysis has been added that considers how our results change when three inputs are varied (ice model resolution, lateral extent of the low viscosity region, depth extent of low viscosity region). These results are presented in Figs 6, S4 & S5). In short, our results are sensitive to all of these aspects, indicating that they should be defined as accurately as possible in any future studies that seek to quantify future RSL changes in the three low viscosity regions we considered.*

4) "A discussion of Point 3, would be great, but this might be bejond this study. But in this case they should relate to such a future extension. They should also compare their results to a 1D ve earth model in order to prove the reliability to compare their 3D results with those of an elastic model."

*Since our primary aim was to determine the large-scale influence of low viscosity areas in producing sea-level fingerprints, we adopted relatively simplistic models of the viscosity structure in RGI regions 1, 2 & 17. Our study was not originally intended as a near-field analysis but went in this direction due to the small influence of the localised viscosity structure on global RSL patterns. We have indicated in the Results and Conclusions that more realistic estimates of regional viscosity structure are warranted based on the results of our sensitivity study.*

*In addition to the results in Fig. 3, we have also added RSL curves based on a 1-D viscoelastic model to Fig. 5 to illustrate that when more typical viscosity values are used, the elastic model gives results within ~1 cm or less of the viscoelastic model even in near-field regions.*

5) "Regarding the setup of the manuscript, I must confess that I am not a native speaker. Nevertheless, I had the impressions, that some statements can be expressed more precisely and, at some places, the discussion can be sharpened."

*Many revisions to the text have been made to accommodate the suggestions and criticisms detailed in the appended pdf document. We thank Dr. Klemann for taking the time to provide such detailed feedback. We agree with most of the suggested changes and so have implemented them.*

**Reviewer 2**

1) "Apart from a few minor comments listed below, I have only one somehow major concern. As the authors explicitly discuss, one of the main differences with respect to an elastic model is the presence of side lobes (or peripheral budges) that move vertically in opposite direction with respect to the areas directly below the ice load. Even though those lobes can only be expected to appear next to the uplifting region, while comparing Figure 4b with Figure 2a I cannot help but noticing that the shape of the subsiding region closely resembles the shape of the low-viscosity region. That makes me wonder how Figure 4 and 5 would look like, if the low-viscosity region were wider (which I think is a plausible assumption). I understand that the main purpose of this paper is a proof concept, and the authors already discuss many limitations of their study. Nonetheless, I think that especially the results shown in Figure 5 are very specific and should better be supported by some minimal sensitivity study. A relatively simple option would be to model an end member, i.e., a 1D-model with the same vertical stratification as the center of the low-viscosity zone. That would provide a reasonable lower bound to the effect of local viscosity structures."

*As described above, a small sensitivity study has been added with the results shown in Figs 6, S4 & S5. One aspect of this addition involved extending (laterally) the region with low viscosity material in the shallow mantle (Fig. S3). The high sensitivity of our results to this aspect of the Earth model (as well as the depth to the based of the low viscosity region), indicates that more realistic and better constrained estimates of this 3-D viscous structure will be an important target for future work. This point is now made in several parts of the revised manuscript (including Abstract and Conclusions).*

Minor comments:

*These have all been addressed.*

"l.11: my -> by (typo).

l.46: mantle -> shallow mantle.

l.53: could you add a short explanation of why plate subduction would change the viscosity structure? It might not be obvious to all readers.

l.62: the sentence is redundant, if layer thickness is already mentioned in the previous line.

l.86: is it 2100 CE or 2090 CE? The caption of Figure 1 seems to suggest the latter.

l.217-218: I wonder if it is realistic not to differentiate between asthenosphere and upper mantle. I understand that this is the model obtained in the reference paper, still it seems a little effort to introduce some reasonable discontinuity."

*As discussed above, we have now included a model run that does isolate the low viscosity zone to a shallow asthenosphere layer (with lower boundary at 150 km depth).*

"Figure 4 seems to have a wrong caption (the panel labels in the text do not match the actual panels in the figure).

Figure 5: in all panels, please add a thin horizontal line to indicate RSL=0."

**Reviewer 3 (Dr. Freymueller)**

1) "The biggest question I have raised by the discussion in lines 235-247. When does the model computation start, and does it include in any way the viscoelastic effect of the 20th century ice loss? It is not explicitly stated, but it appears that the model run starts in 2010, and I suspect that the assumption is that there was no mass change before2010 (again, this was not explicitly stated, but is implied). That is actually a substantial problem for the featured areas of Alaska and Patagonia, because the viscoelastic effects of the 20th century ice loss will be substantial."

*We agree that the pre-2010 loading (particularly that during the 20$^{th}$ century) will be important, as outlined in lines 262-266 of the original submission. We chose not to include this signal in our original analysis as sea-level fingerprints, by convention, define the spatial pattern of sea-level change to contemporary (in this case future) ice mass loss. The signal to past loading changes are included in the so-called 'GIA' component, which is a separate analysis (even though the underlying processes are the same). However, this distinction is often not clearly stated and we agree that our study is a good example of a case where it would be logical to combine past and future loading. One issue in doing this, however, is the large uncertainly in defining 20$^{th}$ century loading changes in this region.*

"I don't think that this invalidates the comparison between the 1D and 3D models in general – if anything, the ongoing viscoelastic response to the 20th century changes will amplify the differences in the near field – and I don't think that this will impact their conclusion that the impacts of the low viscosity regions are essentially local. But it will complicate the discussion that centers around Figure 5. In particular, the position of the hinge line will be affected, so while the trends in the hinge line migration will be largely accurate (especially in the later time periods of the model when the pre-2010 changes are less important as they will be a few relaxation times in the past), any statements about the position of the hinge line may not be accurate without an adequate model for the pre-2010 changes. In particular, the statements about the changes in sign of the RSL trend at Sitka and to a lesser extent Vancouver, Victoria and Rio Grande may or may not be correct here – they depend on the magnitude and sign of the response to the 20th century loss. I don't see any easy way to deal with this aside from re-running all of the 3D models (for the 1D models the viscosity is so high these changes hardly matter). They could incorporate information from previous modeling studies for Alaska and Patagonia, but those model predictions are only available for the present and thus only define the trend at 2010, not the response over the whole 2010-2100 interval). As a first order approximation it may be good enough to run the 2010-2100 model forward in time to 2210 (with no further changes after 2100), and then add a re-scaled and time-shifted version of this to approximate the 20th century effects. It may also be enough to avoid saying "transitions from a fall to a rise", and instead refer to more positive and more negative RSL rates."

*We thank Dr. Freymueller for his suggestion. However, we have decided to take a different approach which, although still crude as it assumes no changes in lateral ice extent, has been applied in past analyses. The approach to defining the 20$^{th}$ century changes is the same as*

*that we applied for the 21$^{st}$ century except that we assume the lateral extent of the ice is fixed and equal to that in the RGI (5.0) database. Thus, at each time step (10 year increments), there is a uniform change in ice thickness that is calculated based on this extent and the volume changes given in Marzeion et al. (2015). Details are provided in the revised manuscript (Section 2.1).*

*Again, since the focus of our work is the impact of low viscosity regions on computing fingerprints (associated with future ice mass loss) considering the GIA signal due to past loading changes is not strictly within our remit. However, by including a 20$^{th}$ century signal we demonstrate the importance of glacier changes during this period on the GIA contribution to future sea-level change in these regions. The results obtained from our loading model are shown in Figs. S6 & S7 and they indicate that, as noted by Dr. Freymueller, the signal is important and so should be included to make accurate RSL projections in these near-field regions.*

2) "Section 2.1, Ice Model. Can you add something about how this 21st century load differs from the 20th century changes? This should be feasible in the case of Alaska (e.g., Berthier et al., 2010, Nature Geosciences), and Patagonia, which should be sufficient examples. For the world as a whole, the total projected mass loss can be compared with existing global estimates from mountain glaciers. I think it would be adequate to give a percentage change for each region – there will be finer-scale spatial detail changes but they are not important here."

*We now include, in Section 2.1, ice volume changes (barystatic values) for the three RGI regions considered. These are taken directly from Huss and Hock (2015). The 20$^{th}$ century loading model is introduced also now described in this section and we give barystatic values for each RGI region (1, 2 & 17) for 1902-2010, as defined in Marzeion et al. (2015).*

Minor points

*Note that the minor points (e.g. misspellings) made below have been addressed even though this is not explicitly stated.*

"Abstract, line 17. The differences 1 cm and 10s of cm don't mean very much without the context of the size of the signal, especially given that the ice load model is a future projection. I suggest at a minimum comparing these to the barystatic sea level rise (10.8 cm) for better context."

*Agree - change made.*

"Line 39. Add "assumed to be" before "dominantly elastic". This assumption is fine in many cases, although not in the examples this paper focuses upon."

*Change made.*

"Line 52. Spelling correction: Freymueller

Line 62. This sentence largely repeats the previous one."

***Sentence deleted.***

"Lines 163-168. Same comment as on the Abstract above. In the case of the far-field regions, I wonder if "<1 cm" is the best way to describe it, as it looks on the figure like «1 cm might apply for large areas. I suggest computing a simple statistic like the median absolute deviation for the regions that they define as far-field in panel c, and reporting this quantitative measure."

***We performed some statistics on the results shown in Fig. 3c and found a mean value (when excluding low viscosity regions) of ~1 mm with a standard deviation of ~3 mm (at 2100 CE). While we do not report the standard deviation, the 1 mm value is now used in the Abstract and Conclusions.***

Line 192. While "greater" and "less than" are mathematically correct, I suggest using the wording "more negative" and "more positive" instead. A similar wording is used a few sentences later. One reason to suggest this change is that it discourages a potential misinterpretation that a reader might make about the absolute value of the RSL change. (I suggest this because my own brain briefly interpreted "less than" as implying "closer to zero" before I caught myself and realized that the signal in question was already negative.

***This paragraph has been extensively re-worded to improve clarity.***

"Line 207. Delete the comma after "change"

Line 265. Spelling correction: Hu, not Huy

Line 274. Spelling correction: van der Wal

Line 279. Do you mean "viscoelastic" rather than "viscosity elastic"?

Lines 287-291. Clarify here whether you incorporated the 20th century changes in your models. My reading is that you did not."

***Done. See response to main comment #1.***

"Figure 1. It is really hard to see the colors on the thickness changes. I suggest using a much thinner black line for the coastline, and/or less detail applied for the coastline. Make sure that the resolution of the figures is good enough that readers can zoom in and see the thickness change projections."

***Suggestion implemented.***

"Figure 2. The regions are a bit arbitrary here, but I think they are basically defensible. I infer from the text that this was a (former?) students' project and thus not likely to easily tweaked, but I don't see any clear reason to have a region of high viscosity between the red and green regions in Figure 2a. This will only matter locally and does not impact the main conclusions, so I don't object if it is not easy to change this and it is left as it is."

*While we agree that making two distinct viscosity regions for RGI regions 1 & 2 is awkward given their proximity and common tectonic environment, we did this because the viscosities inferred in these areas are different. A sentence has been added to make this point.*

"Figure 5. Some of these curves will be substantially affected by the inclusion of the 20th century ice loss, while others will have only minor impacts. See comments above."

*See response to main comment #1.*